# Psychological Impact of COVID-19 Lockdown and Its Evolution: A Case Study Based on Internet Searching Data during the Lockdown of Wuhan 2020 and Shanghai 2022

**DOI:** 10.3390/healthcare11030289

**Published:** 2023-01-17

**Authors:** Wenyuan Zhou, Xiaoqi Zhang, Yanqiao Zheng, Tutiantian Gao, Xiaobei Liu, Han Liang

**Affiliations:** 1Dong Furen Institute of Economic and Social Development, Wuhan University, Wuhan 430072, China; 2Institute of Economics, Chinese Academy of Social Sciences, Beijing 100836, China; 3School of Economics and Management, Southeast University, Nanjing 210096, China

**Keywords:** mental health, COVID-19 lockdown, Baidu search index, depression, domestic violence, evolution

## Abstract

It has been three years since the initial outbreak of COVID-19 in Wuhan, China, which incurred huge damage both physically and psychologically on human’s normal life. As a prevention measure, the lockdown was first adopted by Wuhan, then by a long list of Chinese cities and many other major cities around the world. Lockdown is the most restrictive social distancing strategy, turning out effective in mitigating the spreading of COVID-19 on the community level, which, however, cuts off all social interactions and isolates healthy people from each other. The isolated nature of the lockdown could induce severe mental health issues, forming one major source of depression and domestic violence. Given the potential side effect, a comprehensive investigation based on reliable data sources is needed to evaluate the real psychological impact of COVID-19 lockdown and its evolution over time, particularly in the time when the Omicron variant, known for its low death risk, dominates the pandemic. Based on the Baidu Searching Index data collected for Wuhan and Shanghai, two major cities in China that suffered from long-lasting (over two months) lockdowns in 2020 and 2022, respectively, it is found that the major psychological issue during the lockdown period is not induced by the spreading of COVID-19, but by the execution of lockdown. With the deepening of knowledge about COVID-19 and the decrease in the death risk, the psychological impact of lockdown keeps increasing, while the impact of virus spreading becomes less important and even irrelevant to depression and domestic violence issues. The findings reveal that from the psychological perspective, the negative effect of lockdown already overweighs the positive one, which is especially true for the Omicron variant provided its almost ignorable death risk. Therefore, it is necessary to re-evaluate the yield and cost of lockdown for those countries where the COVID-19 pandemic has not yet come to an end.

## 1. Introduction

It has been three years since the initial outbreak of COVID-19 in Wuhan, in 2020. In the early time of the COVID-19 pandemic, to prevent the virus from spreading, the local government of Wuhan adopted the lockdown, which ordered all residents to stay at home, shut down the intra- and inter-city transportation, and closed all public facilities, including schools, business buildings, restaurants, shopping centers and the like [1]. The lockdown effectively reduces physical contact among people, which cut off the channels of virus spreading [1,2,3]. However, on the other hand, during the lockdown period that lasted for more than two months from 23 January to 18 April 2020, people were not allowed to walk out of their living community, nor to participate in any kind of off-line social activity, which, therefore, isolated people from their normal social life and increased the risk of mental health problems, including but not limited to depression [4,5,6], insomnia [4,5,6,7,8,9], suicide intention [4,10,11,12], stress [13,14], anxiety [8,10], sadness [10,11] and domestic violence [4,10,11]. 

Despite the extensive studies that documented the stylized fact that the reports of mental health issues increased significantly during and after the months-long lockdown, it remains questionable whether the reported psychological problem should attribute to the lockdown, or alternatively, to the spreading of COVID-19. In fact, the impact of COVID-19 on mental health happened via a complex process. For instance, some studies predicting the psychological distress related to COVID-19 [15] indicated that during the first wave of COVID-19 living with others was a risk factor for higher psychological distress, likely in the relation to the fear of contagion. On the other hand, there were also researches that revealed the contrary [16] that hope has a higher protective role on the psychological distress related to the COVID-19 pandemic for people living with others, indicating the protective role of social connectedness. Due to the existence of the complex mechanisms, it is not fair to simply attribute all the impact on mental health observed during the COVID-19 pandemic and the lockdown period to the lockdown, nor to the pandemic itself. On the other hand, as the lockdown is always a hysteresis response to the severe outbreak of the pandemic [1,2], while the end of lockdown is also accompanied by the end of the pandemic, it is not easy to disentangle the effect of lockdown from the total effect of the pandemic. In the terminology of statistics, there exists the identification issue of the psychological impact of lockdown, which is critical to policymakers as it affects the way to evaluate the cost of lockdown.

In addition to the identification problem, most of the existing studies on the mental health impact of COVID-19 rely heavily on cross-sectional survey data [17,18,19]. As always being criticized, cross-sectional survey data has a limited sample size and limited reachability to certain groups of people which makes it less likely to be representative of the full population. In addition, the data quality is sensitive to the design of survey questions and the attitude of interviewees. Given these concerns, the findings based on survey data might be biased.

To these problems, the study aims at providing a new investigation into the psychological impact of the COVID-19 lockdown via the Baidu Searching Index (BSI) data, which is behavioral data of internet searching happened on the biggest search engine in China (Baidu.com, https://index.baidu.com/v2/index.html#/ access date 7 October 2022). Internet searching data has been widely applied in literature as an efficient indicator for public opinion, mental status, and social trends [20,21,22,23,24,25]. In China, it is even more efficient than anywhere else to take the BSI data extracted on the base of medical- and health-related searching as a proxy for the real health status of the full population. This is because China has the broadest range of Internet users in the world. According to the 49th China Statistical Report on Internet Development, there were a total of 1.032 billion netizens as of 31 December 2021. The report also claimed that more and more people use the Internet to search for health-related information. In addition, according to the search engine market report published by the renowned market research agency, StatCounter, at the end of 2021, 85.48% of the netizens in China used the Baidu platform to search for information as of 31 December 2021. See https://gs.statcounter.com/search-engine-market-share/all/china (accessed on 7 November 2022). Baidu is the largest Chinese-language search engine and one of the ten largest global websites. Baidu offers many services, including searching for websites, maps, videos, news, pictures, encyclopedia, multimedia files, translation, and other useful functions, as well as the mobile service Baidu App. Existing research has demonstrated that the availability of reliable health information for patients is of great importance; it can significantly increase self-efficacy, reduce anxiety, and enhance self-care ability [26,27]. Notably, because of the low costs and rich resources, the Internet has become a major source for seeking health-related information, such as providing options on how to access health services and professional advice [28,29]. Therefore, compared with the survey data, the search data is more representative of the full population given that there is no accessibility restriction to the internet for every subgroup of the population [20,30,31].

As for health-related searching, Zhang et al. [32] find that 71.79% of Chinese adults have received health education on the Internet, and almost all (98.35%) had searched online for health-related information. They further find that Baidu is one of the most popular tools for seeking health-related information in China. The wide-range use of Baidu health searching guarantees the generic representativeness of the BSI data to the real mental-health status of the full population. In addition to the population representativeness, online health-related information is especially suitable for internet users because of the low cost and rich resources [33,34], which makes the Internet an alternative source of help in times of mental illness, especially for those with internalized stigma [35]. Compared to survey data, the BSI data is collected from the active searching behavior of internet users, rather than from the passive answering of the interviewees, which makes BSI data not suffer as much from the honest issue as the survey data as the interviewee may not be willing to honestly answer the question being asked [36,37]. The cross-sectional survey data also suffers from the low-frequency issue [30,37,38], because the survey cannot be done every day, which makes it impossible to monitor the mental health status change of some certain groups of people in a timely manner. However, the impact of lockdown on people’s mental health is a gradually evolving process, and it turns out that this dynamic nature is important to help us disentangle the effect of lockdown from that of the COVID-19 pandemic [39]. In contrast to the cross-sectional survey data, internet search data can be collected every day which provides us with a powerful tool to capture the dynamic change in people’s mental health status along with the persisting lockdown [5]. Given the aforementioned advantages, when confronted with depression, compared with other available sources, the Internet may be a more proper source of information for people with internalized stigma and limited access to health care services [40].

On the base of BSI data, we target our analysis to Wuhan and Shanghai, two major cities in China that experienced months-long lockdowns in 2020 and 2022 respectively. It is worth to note that the lockdown happened to the two cities on two different stages of the COVID-19 pandemic. For Wuhan, it was locked down in the earliest stage of the pandemic when the death rate of the virus was relatively high. In contrast, Shanghai was locked down during the pandemic of the Omicron variant of COVID-19, which is renowned for its high infection rate but low death risk. The excessively high infection speed makes the lockdown less effective in preventing the spreading of COVID-19 [41,42], while the less risky nature of Omicron makes the necessity of lockdown discountable [43,44,45,46]. Due to the different backgrounds of the lockdown of Wuhan and Shang, it is expected that their impact on people’s mental health should be different. Theoretically, the reduced death risk of Omicron variants should reduce the anxiety and panic incurred purely from the epidemic spreading, which should offset the negative impact of the COVID-19 pandemic on mental health. In the meantime, the degree of the negative impact of the execution of strict lockdown on mental health should increase at least in relative to the impact of the pandemic itself. If this hypothesis held, it would also support the argument that the lockdown, as a preventive measure to the pandemic, is enough on its own to generate severe side effects on people’s mental health. As to test the validity of this hypothesis, we will make a comparative analysis between the lockdown of Wuhan and Shanghai. We highlight that this comparison is meaningful both in practice and to the literature, it does not only improve our understanding of the net effect of lockdown on mental health and its evolution over time in alignment with the mutation of COVID-19, but is also critical to the yield-cost evaluation on the prevention strategies of COVID-19 pandemic for the countries to which the COVID-19 pandemic have not come to an end.

## 2. Materials and Methods

In this section, we describe how the data is collected and pre-processed. On that base, we present a brief overview of the event study approach and the DID regression analysis that will be applied to filter the impact of the lockdown, out of the total effect of COVID-19, on people’s mental health.

### 2.1. Baidu Search Index Data

Due to the dominant position of Baidu company in the Chinese searching market and the availability of the search index data, we select Baidu Search Index (BSI) as the data source for this study. Similar to Google Trends, BSI is calculated as the weighted sum of the search frequency of each keyword via the platform run by Baidu, including both the web and the mobile terminal [4]. We extract Baidu Search Index on dates from 1 January 2019 to 31 August 2022, covering the time of the two lockdowns (23 February to 7 May 2020 for Wuhan and 28 March to 1 June 2022 for Shanghai). Data in 2019 are included to control for the seasonality trend for the Wuhan lockdown, and Data in 2021 are included to control for the seasonality trend for the Shanghai lockdown. The summary features of the BSI time series will be reported in the next section.

To capture the overall mental health status of residents in a city, we follow the literature [4,5] to collect the BSI for a set of mental-health-related keywords. For completeness, we collect all high-frequent used mental-health related keywords in literature [4,5,6,7,8,9,10,11,12,13,14], which include “depress”, “boredom”, “anxiety”, “stress”, “unhappiness”, “sadness”, “sleepless”, “domestic violence”, “verbal violence”, “conjugal violence”, “suicide” and “self-harm”. To simplify the analysis, we group these keywords into two big classes, which are depression and domestic violence. For depression, the keywords include “depress”, “boredom”, “anxiety”, “stress”, “unhappiness”, “sadness” and “sleepless”; while for domestic violence, the keywords include “domestic violence”, “verbal violence”, “conjugal violence”, “suicide” and “self-harm”. The BSI for each topic is the mean of the BSI for each keyword associated with that topic. As the BSI is counted on the search frequency of the mental-health-related keywords within a city, the greater BSI value implies the worse mental health status on the aggregated level for the target city. By the BSI data collected on a daily base for the topic of depression and domestic violence, we apply the event study and the DID regression to filter the impact of lockdown out of the total impact of COVID-19.

### 2.2. Event Study

The event study is essentially a regression model [11] which, in the context of the current study, can be specified as follows:(1)$$M_{i,c}=\overset{12}{\underset{k=-2}{\sum }}\alpha_{k}E_{k}*Year_{i}*City_{c}+\overset{12}{\underset{k=-2}{\sum }}\beta_{k}E_{k}+\gamma X_{i-1,c}+\mu_{i}+\rho_{c}+\epsilon_{i,c}$$
where $M_{i,c}$ represents the BSI for the given topic on day i in city c, $Year_{i}$ is the dummy for the year of the lockdown, which corresponds to 2020 in the Wuhan Lockdown and 2022 to the Shanghai Lockdown, $City_{c}$ is the dummy variable that equals one for the city of Wuhan (or Shanghai) during the Wuhan (or Shanghai) Lockdown. The model includes year, week and day (Monday to Sunday) fixed effects that appear in the vector μ. $E_{k}$’s are dummy variables for the three weeks before the start of lockdown and the twelve weeks after the start of the lockdown. The third week before the start of the lockdown (in 2019 and 2020) is the reference period. The standard errors are robust and are clustered at the day level. Regressions according to Equation (1) are carried out for the Shanghai lockdown (2022) and Wuhan lockdown (2020) separately.

Behind the regression Equation (1) is the key assumption that when considering the effect of the Wuhan lockdown in 2020, we take Shanghai’s sample collected at the same time as the control group, because during the Wuhan lockdown, there do exist epidemic spreading in Shanghai but no as strict lockdown being executed as in Wuhan. While considering the effect of the Shanghai lockdown in 2022, we take Wuhan’s sample collected at the same time as the control group due to the same reason. Taking the other city as the control group during a city being locked down is the key to disentangling the psychological impact of lockdown from the total impact of COVID-19, which facilitates us to compare the compound effect of lockdown plus the COVID-19 spreading with the pure effect coming from the COVID-19 spreading only. The pure effect of epidemic spreading is exactly what is to be captured by adding the control group.

Another assumption behind Equation (1) is that in the absence of a lockdown, BSI would have evolved in the same way as in the year prior to the lockdown. Based on this assumption, we are able to compare the difference of the differences in search intensity for the same topic within the weeks that are pre- and during- the same lockdown periods between two consecutive years.

Despite that taking Shanghai/Wuhan as the control group for each other during the lockdown year can help filter out the psychological impact of lockdown, it is undeniable that when Wuhan/Shanghai being locked down, the spreading of COVID-19 in the other city is not as severe as that in the lockdown city. The difference in the severity might induce bias in the statistic inference. To fix this problem, we add the control variable $X_{i-1}$ to neutralize the severity issue with $X_{i-1}$ denoting the logarithm of the 1-day lagged number of newly infected cases of COVID-19 in the city under study. Given that, the regression coefficients $\alpha_{k}$s should reflect the net impact of lockdown on BSI for a given topic at the k-th week pre- and during-lockdown. As the BSI reflects the search frequency of the related mental health issues, the greater of the BSI value implies a more severe mental health problem for the target population. Therefore, if the lockdowns do have a negative impact on the mental health status, a significant positive sign can be expected for the $\alpha_{k}$s when *k* > 0.

To further validate that there is no such effect that was induced by COVID-19 spreading rather than lockdown being mixed in $\alpha_{k}$s, we will make a robustness check for the trend of the $\alpha_{k}$s and the weekly trend of the aggregated infectious cases. The $\alpha_{k}$s does capture the net psychological impact of lockdown only If the two trends are not parallel to each other.

### 2.3. DID Regression

Despite that taking Shanghai/Wuhan as the control group for each other is helpful to isolate the effect of the lockdown from that of COVID-19 spreading, it barricades the cross-period comparison between the Wuhan lockdown in 2020 and the Shanghai lockdown in 2022. As highlighted in the introduction, the virus being spreading in Wuhan 2020 and that in Shanghai 2022 is significantly different in their death risk [47,48]. In addition, with the experience of the COVID-19 pandemic for two years, the public’s attitude and knowledge toward the virus changed significantly as well [43]. These changes should have made COVID-19 much less fearful for the public in 2022 than in 2020 [43]. The reduced threatening should decrease the negative psychological impact of the spreading of COVID-19, which can be witnessed from the cross-period comparison between the Wuhan and Shanghai Lockdown. To make this comparison feasible, we shall adopt the following DID analysis based on the transformed BSI data.

Similar to the event study, the DID analysis is based on a regression model [4] that can be specified according to the setting of this study as below:(2)$$M_{i}^{\prime }=\alpha^{\prime \;}CY_{i}*During_{i}+\beta_{D}^{\prime }During_{i}+\beta_{C}^{\prime }CY_{i}+\gamma \prime X_{i-1}+\mu_{i}^{\prime }+\epsilon_{i}$$
where $M_{i}^{\prime }$ is the transformed BSI data for a given searching topic on the day *i* with *i* being a date either in 2020 or in 2022, which is derived from the original BSI data in the following way:(3)Mi′=Mi,WuhanMi,ShanghaiMi−365,WuhanMi−365,Shanghai, if i represents a date in 2020; Mi,ShanghaiMi,WuhanMi−365,ShanghaiMi−365,Wuhan, if i represents a date in 2022.

In Equation (3), the transformed BSI $M_{i}^{\prime }$ is essentially a double ratio of the original BSI, the inside ratio $M_{1,i}=\frac{M_{i,c}}{M_{i,c\prime }}$ uses the BSI for the control-group c′ to rescale the BSI for the treatment-group city c. By doing so, we can annihilate the city-level difference behind the BSI. The outside ratio $\frac{M_{1,i}}{M_{1,i-365}}$ is the year-over-year change of the inside ratio which captures the potential impact induced by lockdown given that there was no strict lockdown being executed for both cities in the year 2019 and 2021. From the construction of the transformed BSI in Equation (3), $M_{i}^{\prime }$ for *i* being a day in 2020 reflects the percentage of the search index change of Wuhan in relative to Shanghai on day *i* versus that on the same day one year ago, while for *i* being a day in 2022 reflects the percentage of the search index change of Shanghai in relative to Wuhan on day *i* versus that on the same day one year ago. Under this interpretation, $M_{i}^{\prime }$ for *i* being a day in 2020 can be thought of as the BSI for the city-year pair of Wuhan and 2020, while for *i* being a day in 2022 be thought of as the BSI for the city-year pair of Shanghai and 2022.

Given the transformed BSI, we take the Wuhan-2020 pair of samples as the control group and Shanghai-2022 pair of samples as the treatment group. In the other words, the samples are selected into treatment group according to their experience with the Omicron variant. We shall use the dummy variable $CY_{i}$ to indicate the membership of treatment group. The dummy variable $During_{i}$ indicates whether the day *i* is within the lockdown period. Under all these specifications, the key coefficient $\beta_{D}^{\prime \;}$ captures the total effects of lockdown, $\beta_{C}^{\prime \;}$ stands for the effect of the long lasting of COVID-19 pandemic into the Omicron stage, while $\alpha^{\prime \;}$ captures the effect of lockdown on the Omicron stage.

Given that the death risk of COVID-19 reduces significantly over the Omicron stage, it is expected that the difference in the impact of Omicron variant and the original COVID-19 on mental health, $\beta_{C}^{\prime \;}$, should be negative. On the other hand, if lockdown is a major source of the mental health issue, a positive sign for the coefficient $\beta_{D}^{\prime \;}$ can be expected. Furthermore, if the negative psychological impact of lockdown was strengthened during the Omicron stage, a significant negative sign for $\alpha^{\prime \;}$ should be witnessed. The expected signs of regression coefficients in Equation (2) and their theoretical implications are summarized in Table 1.

## 3. Results

### 3.1. Graphic Summary for Baidu Searching Index

In this section, we report the global features of the trend of the BSI time series. To be comparative, for each sample city, Wuhan and Shanghai, and each topic, domestic violence, and depression, we separate the time series over 2019–2022 into two periods. The first period spans from 1 January to 30 June within both 2019 and 2020, which includes the lockdown period of Wuhan inside. The samples in 2019 are collected as the benchmark of comparison. The second period spans from 1 January to 31 July within both 2021 and 2022, which includes the lockdown period of Shanghai inside. The samples in 2021 are collected as the benchmark of comparison. We will report the BSI time series within each of the two periods separately. The time series within each period is cut into two pieces according to their occurrence year, and the two sub-series are aligned together according to the occurrence date in the corresponding year.

The BSI series for the topic of domestic violence (depression) is reported in Figure 1 (Figure 2) against the occurrence period (2019–2020 in the first line, 2021–2022 in the second) and the city (Shanghai associates with the left side, Wuhan with the right side). As we only concern with the BSI series in a time interval including the lockdown period of Wuhan and Shanghai, instead of the whole-year-long data, we shall truncate the time series in Figure 1 and Figure 2 to make the report concentrative. The two vertical red lines within each sub-figures of Figure 1 and Figure 2 represent the start date and end date for the lockdown of Wuhan (2020) and Shanghai (2022).

According to Figure 1, the intensity of searching for domestic-violence-related topics by terminal IPs registered in Wuhan tended to be above what occurred in one year ago, at least within the first month since the Wuhan lockdown. In contrast, during the same period, the search intensity for Shanghai IPs was not much different from that in one year ago. This observation suggests that as the epicenter, the severe shock of the COVID-19 epidemic crisis in Wuhan and its incurred lockdown did generate significant negative impact on residents’ mental health.

During the Shanghai lockdown, Figure 1 shows that the intensity of searching for domestic violence by Shanghai IPs was almost uniformly above that during the same time period in one year ago, which, notably, occurred exactly within the lockdown period, neither before nor after. The same pattern does not hold for Wuhan which did not experience any kind of lockdown during the lockdown period in Shanghai. Provided that the infectious cases in Shanghai had already started to sharply increase two weeks before the start of the Shanghai lockdown (see the analysis in the next section), the observation above suggests a mismatch between the time point when the infectious cases start to increase and that when the searching for domestic violence started to increase, which further suggests that the lockdown is more likely than the sharply increased infectious cases to be the source of the increasingly searching for domestic violence.

In Figure 2, compared with one year ahead, the intensity of searching for depression-related topics did not show any distinct pattern during Wuhan lockdown for both Wuhan and Shanghai. During the Shanghai lockdown, the depression-related search did not show much difference compared with that one year ago for Wuhan, but it uniformly increased for Shanghai. In particular, the increasing pattern was preserved even after the end of the lockdown. This observation is also consistent with the findings in the literature that strict lockdown would promote strong resistant motives which enforced people to be depressed and the depressed mood can persist even after the disappearance of its origin [4]. Beyond the literature, our findings also suggest that the severity of the mental-health impact of lockdown tends to increase, facing the low death risk of the Omicron variant.

Based on the brief overview of the BSI time series, positive evidence was already found to support our hypothesis that it is the lockdown, instead of the outbreak of COVID-19, that generates a negative impact on people’s mental health, especially when the virus becomes little harmful to human’s life. To examine this hypothesis more rigorously, we will conducte a more statistical analysis on the base of BSI data in the next sections.

### 3.2. Results for Event Study

In this section, the event study approach is applied to the BSI time series. According to Equation (1), for each event week k, the psychological impact of lockdown for people in the treatment group city is measured by $\alpha_{k}$, while the impact for people in the control group city is measured by $\beta_{k}$ only. In Figure 3 (Figure 4), we report the trend of these impact measures and their statistical significance for the topic of domestic violence (depression) in both cities. In the figures, the error bar is plotted according to the length of the 5% confidential band. To compare the dynamic trend, the daily increment of infectious cases during the same period for each city is also plotted in the corresponding subfigure of Figure 3 and Figure 4.

According to Figure 3, the impact of the Wuhan lockdown on domestic violence for families in Wuhan is not statistically significant, reflecting as the horizontal zero line is always contained within the 5% confidential bands for all 12 event weeks in the study. In contrast, a significant impact on the domestic violence search by Shanghai IPs is witnessed in the second week after Wuhan started lockdown. This observation can be interpreted as a consequence of the asymmetricity between the limited infectious cases and the rigorous restriction on social contact in Shanghai during the Wuhan lockdown. In fact, Shanghai was partially locked down during the time of the Wuhan lockdown, despite that Wuhan was the only epicenter of COVID-19 at that time and the infectious cases in Shanghai were very limited [1,3]. The asymmetricity between the limited infected cases and the rigorous prevention policy in Shanghai could incur bad moods and quarrels between family members which further stimulated domestic violence. In contrast, in the epicenter, Wuhan, the outbreak of COVID-19 had caused a large number of death cases which made the public believe in the worthiness of the lockdown and induced fewer emotional quarrels and violent events.

While Shanghai was locked down, a significant increase in the search frequency for domestic violence by Shanghai IPs is witnessed since the second week, which lasted until the 7th week after the lockdown was initiated. Comparatively, the Shanghai lockdown did not entail a statistically significant psychological impact on the searching behavior of Wuhan IPs except for the second week after the initiation of the lockdown. This observation also agrees with that in Figure 2. For week 2, despite its significance, the scale of the coefficient $\beta_{2}$ is relatively small in contrast to $\alpha_{2}$, which does not account for much meaningful implication.

From Figure 4, the Wuhan lockdown did not generate a significant impact on the search for depression by both Wuhan and Shanghai IPs. In contrast, the Shanghai lockdown did significantly increase the intensity of searching for depression by Shanghai local IPs, while it also increased the searching intensity by Wuhan IPs over week 1, week 4, and week 9. This observation basically coincides with that in Figure 2. For the significant increase in the search intensity in Wuhan, it can be explained by the spillover effect of information propagation on the internet. During the Shanghai lockdown, there were a lot of videos and blogs published on social media that self-reported the authors’ bad feelings and panics regarding the lockdown. These contents can be watched by any person from everywhere as long as they have internet access, which makes the depressed mood spread via the internet to the outside world. Detailed discussions on the spreading of information during an epidemic crisis and its impact on mental health can be found tremendously in literature [49,50], we will not expand it in this study. However, as a remark, we highlight that the increased depression-related search in Wuhan incurred by Shanghai lockdown reveals once again that the lockdown might be more powerful than the spreading of COVID-19 in causing mental health issues.

To more rigorously disentangle the impact on mental health incurred by the spreading of COVID-19 and by the lockdown, we add the time series of the daily number of new infectious cases (including the asymptomatic infections) over the 12 event weeks into Figure 3 and Figure 4. From a detailed exam of the infection curve and the trend of psychological impact coefficients, it is found that neither the peak nor the bottom of the infection curve can match the variation trend of the coefficients of $\alpha_{k}/\beta_{k}$s. Especially during the Shanghai lockdown in 2022, for both depression and domestic violence searching, the psychological impact measure $\alpha_{k}$ persisted to be significantly positive and maintained at a relatively high level even after the daily increment of infectious cases had decayed far away from the peak. This observation excludes the argument that the number of COVID-19 infection, the key measure adopted by China’s CDC for the epidemic severity and the decision of lockdown [41], determines the psychological impact of COVID-19 over the pre- and during-lockdown periods. From this perspective, we can conclude that the lockdown on its own could be a major source of the negative impact on people’s mental health.

### 3.3. Results for DID Regression

Based on the event study analysis in the previous section, the lockdown policy on its own is already sufficient to generate a significant negative impact on the mental health of the people being locked down. The COVID-19 pandemic has been persistent for three years, then a natural question to ask is whether the long-lasting nature of the pandemic and the persistent variation of the virus with the decreasing death risk have changed the way the impact of lockdown has on mental health. To answer this question, we apply the DID regression approach to the BSI data, the main results are reported in Table 2.

In Table 2, the regression coefficients for the key variables and their statistical significance are reported against the transformed BSI for the two topics, domestic violence, and depression. It is shown that the coefficients for the treatment group dummy are negative for both topics and significant at a 1% level for the topic of domestic violence and significant at a 5% confidential level for depression. According to the design of the treatment group, which corresponds to the city-year pair of Shanghai and 2022, this result suggests that the decreasing death risk of the virus reduces the impact of COVID-19 on people’s mental health which is consistent with the expectation.

In contrast to the reduced psychological impact of the virus, the significantly positive (at 1% confidential level for both topics) coefficient for the lockdown dummy implies that regardless of the status of COVID-19 spreading, only the execution of lockdown is sufficient to rise up the search frequency 16% and 8.6% for domestic violence and depression, respectively. This result is quite striking. Especially when compared with the coefficient for the treatment group dummy, the absolute value of the coefficient of the lockdown dummy is uniformly larger, which suggests that during the Shanghai lockdown in 2022, the aggregated impact of the COVID-19 pandemic on people’s mental health is still negative significantly. However, compared with the lockdown period of Wuhan in 2020, the lockdown policy contributed much more to the total negative mental impact in 2022, despite that the policy was deliberately designed to protect people from the virus.

This sad argument is even reinforced by the positive sign for the coefficients of the interaction term for both the depression and domestic violence search, which is not statistically significant for domestic violence but extraordinarily significant for depression (at 1% confidential level). For the topic of depression, on the base of the Wuhan lockdown, the search frequency is lifted up for an extra 14.5% during the Shanghai lockdown. This result suggests that compared with Wuhan 2020, the contribution of lockdown to people’s worse mental health status is not only increased relatively (in relative to the contribution of the infection of COVID-19 and its death risk), but also enhanced in the absolute sense, although the death risk of the virus has reduced drastically.

## 4. Conclusions and Discussions

Based on the empirical results from the BSI time series for the search topics of domestic violence and depression, we extract three major findings:

1.It is not the spreading of COVID-19 but the lockdown itself that forms the major source of the negative impact on people’s mental health.2.Compared to the first wave of the COVID-19 pandemic, the contribution of lockdown to the total negative psychological impact of COVID-19 was increased significantly in relative to the negative impact incurred by the COVID-19 spreading, during the Omicron variant became the mainstream strain.3.Beyond the relative contribution, the absolute value of the negative psychological impact coming only from lockdown was also increased significantly in the Omicron age.

The three findings suggest an increasing trend of the side effect of lockdown as the major preventive measure against the COVID-19 pandemic, which gives an alert to the policymakers to re-evaluate the cost and yield of prevention strategies against COVID-19.

The findings also inspired us the rethink the design and effectiveness of the prevention system against the COVID-19 pandemic. In fact, any preventive measure, including the lockdown, is designed to mitigate its negative impact on people’s health, both physically and mentally. On the other hand, every epidemic crisis, including the COVID-19 pandemic, would induce serious sentimental issues, including panic, fear, anxiety and the like, which will lead to depression and/or other kinds of mental illnesses [4,5,6,7,8,9,10,11,12,13,14]. From this perspective, as long as the preventive measures can quickly reduce infectious and dead cases, it would be helpful to cure mental problems as well. This reasoning explains why we did not see that the Wuhan lockdown in the earliest stage of COVID-19 pandemic intrigued the searching for depression- and domestic-violence-related topics. Therefore, in aspect of mental health, at the stage of Wuhan lockdown, lockdown is not that harmful and costly [5,7].

However, as the persistence of the COVID-19 pandemic for a longer time is accompanied by a reduced death risk, our empirical evidence shows that at least in the aspect of mental health, the epidemic spreading becomes less costly, while the lockdown becomes increasingly costly. This conclusion agrees with the recent studies on the Shanghai lockdown [51,52]. Particularly in the time that the Omicron variants dominate, the lockdown has already accounted for the major cause for the intensive searching of depression and domestic violence on the Internet. This fact calls for an urgent re-assessment on the role and the social economic cost of lockdown by the decision-makers in the countries where lockdowns are still thought of as the silver bullet to the COVID-19 pandemic.

There also exist limitations and potential extensions for our study. First, in this study, we only consider the BSI for two sampled cities. Although it is barely sufficient from the perspective of a case study, it could restrict the generality of the conclusion. As a direction of future studies, it would be promising to collect the BSI for all prefecture-level cities in China [4], during 2019–2022, by which we can design subtle control groups to give a more precise filtration on the impact of lockdown. Second, since 2021, the lockdown is more frequently applied by many Chinese cities as an effective preventive measure against the COVID-19 pandemic. The difference in the execution time, frequency, and strictness of lockdowns in different cities of China provide a quasi-natural experimental environment. A promising direction of future studies is to collect the data of the official documents on lockdown for all the prefecture-level cities in China. By matching the lockdown policy data with the BSI data, more advanced analysis can be carried out to dig deeply into the relation between lockdown and people’s mental health.

## Figures and Tables

**Figure 1 healthcare-11-00289-f001:**
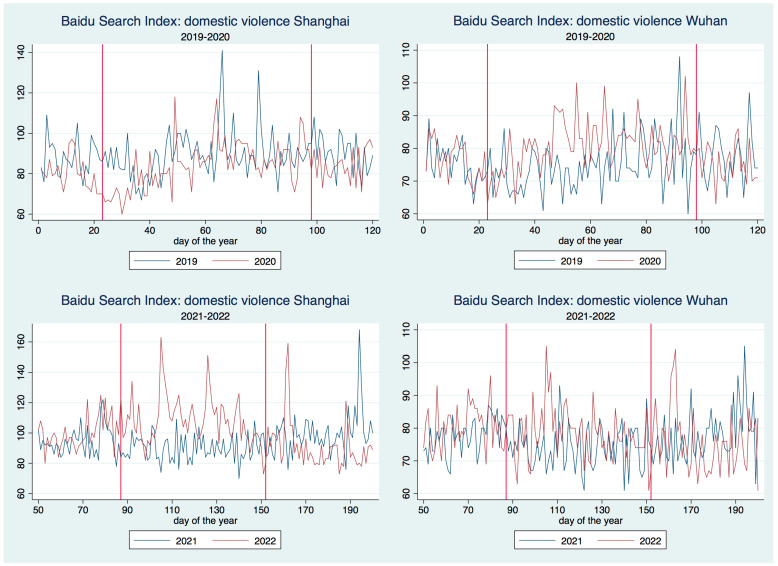
BSI trend for domestic-violence-related topic, Shanghai vs. Wuhan, 2019–2022.

**Figure 2 healthcare-11-00289-f002:**
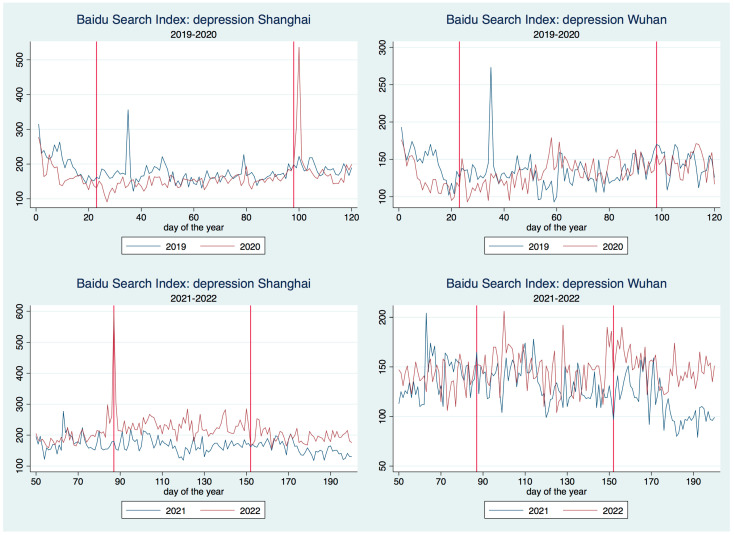
BSI trend for depression-related topic, Shanghai vs. Wuhan, 2019–2022.

**Figure 3 healthcare-11-00289-f003:**
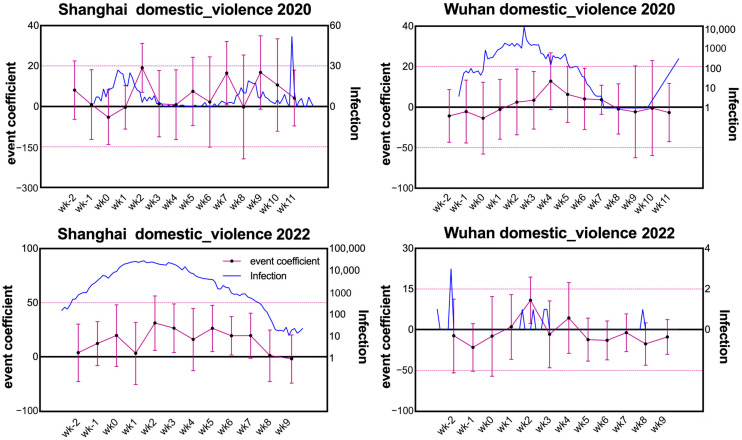
Trend of $\alpha_{k}/\beta_{k}$ for domestic violence, Shanghai vs. Wuhan, 2019–2022.

**Figure 4 healthcare-11-00289-f004:**
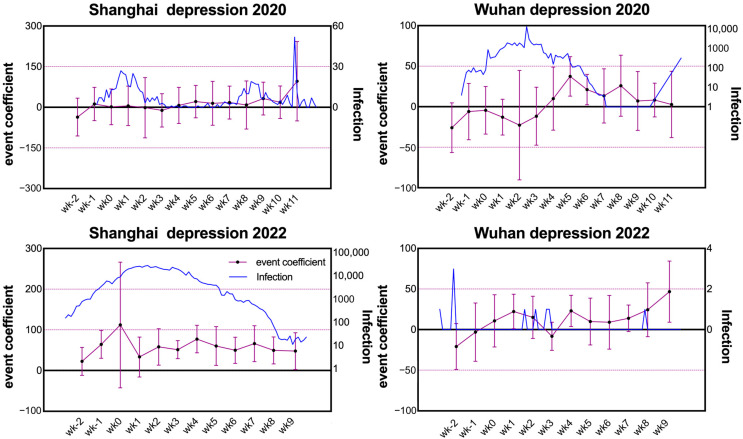
Trend of $\alpha_{k}/\beta_{k}$ for depression, Shanghai vs. Wuhan, 2019–2022.

**Table 1 healthcare-11-00289-t001:** Meanings of the sign of DID regression coefficients.

$\beta_{C}^{\prime \;}$	+	The psychological impact caused by epidemic spreading increase from 2020–2022
	−	The psychological impact caused by epidemic spreading decrease from 2020–2022 (expected)
$\beta_{D}^{\prime \;}$	+	Lockdown has negative shock on mental health (expected)
	−	Lockdown has positive shock on mental health
$\alpha^{\prime \;}$	+	The impact of lockdown on mental health is increased in Omicron stage
	−	The impact of lockdown on mental health is decreased in Omicron stage

**Table 2 healthcare-11-00289-t002:** Estimate to the key coefficients in DID regression Equation (2).

Variables	Domestic Violence	Depression
$CY_{i}(\beta_{C}^{\prime \;}$)	−0.054 ***	−0.053 **
	(0.019)	(0.024)
$During_{i}$($\beta_{D}^{\prime \;}$)	0.164 ***	0.086 ***
	(0.031)	(0.028)
$CY_{i}*During_{i}$ (α′)	0.02	0.145 ***
	(0.042)	(0.047)
# Obs.	623	624
$R^{2}$	0.1178	0.1696

Note: (i) * *p* < 0.10, ** *p* < 0.05, *** *p* < 0.01. (ii) The standard deviation of the estimation is reported in (.)

## Data Availability

Publicly available datasets were analyzed in this study. This data can be found in https://index.baidu.com/v2/index.html#/ (accessed on 8 December 2022).

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
