# Peer review of "Psychological Impact of COVID-19 Lockdown and Its Evolution: A Case Study Based on Internet Searching Data during the Lockdown of Wuhan 2020 and Shanghai 2022"

_healthcare, 2023, doi:10.3390/healthcare11030289_

Round 1

Reviewer 1 Report

In the article entitled "Psychological impact of COVID-19 lockdown and its evolution: A case study based on internet searching data during the lockdown of Wuhan 2020 and Shanghai 2022" the authors do an internet search on people's behavior during the lockdown. 
The authors concluded that during the lockdown, and in this case due to the pandemic, domestic violence and depression increase in people.
Studies have already been conducted where they show that people's behavior tends to depression and violence under conditions of confinement.
The authors used a particular search engine of Chinese origin, however it would be interesting to compare it with another search engine such as Google Trends to determine if similar results are obtained or if they are totally different.
The article is well explained and developed and gives us information about the behavior of people in a particular region of China where the pandemic has been the center of the pandemic.

Author Response

Dear reviewer,

Thank you for your comments on our manuscript entitled “Psychological impact of COVID-19 lockdown and its evolution: A case study based on internet searching data during the lockdown of Wuhan 2020 and Shanghai 2022” (ID: healthcare-2120478). Those comments are very helpful for revising and improving our paper. We have studied the comments carefully and made the response as below:

Comment (1):

In the article entitled "Psychological impact of COVID-19 lockdown and its evolution: A case study based on internet searching data during the lockdown of Wuhan 2020 and Shanghai 2022" the authors do an internet search on people's behavior during the lockdown.  The authors concluded that during the lockdown, and in this case due to the pandemic, domestic violence and depression increase in people. Studies have already been conducted where they show that people's behavior tends to depression and violence under conditions of confinement. The authors used a particular search engine of Chinese origin, however it would be interesting to compare it with another search engine such as Google Trends to determine if similar results are obtained or if they are totally different. The article is well explained and developed and gives us information about the behavior of people in a particular region of China where the pandemic has been the center of the pandemic.

Response from the (co-)authors:

Thanks for the comments. As for that we did not compare our findings based on Baidu searching index with Google trend, the main reason was that Google did not provide search engine service within the mainland China due to the political reason. Therefore, we can at most get the google trend based on the searching behavior outside mainland China, this kind of trend data is meaningful on its own, but is irrelevant to the topic of the current study. To this concern, we did not use the Google trend for comparison. We wish the reviewer could understand this problem. Thanks.

Finally, we sincerely appreciate the valuable comments by the reviewer, which help a lot in improving the quality of our paper.

Reviewer 2 Report

Thank you for the opportunity to review the manuscript “Psychological impact of COVID-19 lockdown and its evolution: A case study based on internet searching data during the lockdown of Wuhan 2020 and Shanghai 2022” (healthcare-2120478).

The authors evaluate the psychological impact of COVID-19 lockdown for Wuhan and Shanghai using different searching index data.

This topic is interesting and necessary, but the paper needs to be fundamentally revised.

Clear research question and hypothesis are missing. The methods part is very confusing. for example, under 2.1. channels of health-related information is discussed. Authors should rewrite the introduction an put parts of 2.1 there.

Line 147-158 is enough for 1.2. methods section.

Please combine then the introduction with your results.

This could also strengthen the discussion, as it is quite common to refer to findings from those studies relative to the current study findings in the discussion and conclusions sections.

Conclusions must be drawn. These should contain implications for further research and derive recommendations for action.

Please add a table with the most important findings.

Several Limitations should be listed.

Author Response

Dear reviewer,

Thank you for your comments on our manuscript entitled “Psychological impact of COVID-19 lockdown and its evolution: A case study based on internet searching data during the lockdown of Wuhan 2020 and Shanghai 2022” (ID: healthcare-2120478). Those comments are very helpful for revising and improving our paper. We have studied the comments carefully and made corrections. The main corrections are in the manuscript and the responds to the reviewers’ comments are as follows (the replies are highlighted with red color).

Comment (1):

Clear research question and hypothesis are missing. The methods part is very confusing. for example, under 2.1. channels of health-related information is discussed. Authors should rewrite the introduction as put parts of 2.1 there. Line 147-158 is enough for 2.1. methods section.

Response from the (co-)authors:

Thanks for this revision proposal, we have followed the suggestion to substantially squeeze the section 2.1 and move the discussion regarding the channels between online searching behavioral and mental health status into the introduction. We also add a paragraph in the introduction section to highlight the major hypothesis that the paper attempts to examine and the main findings.

Comment (2):

Please combine then the introduction with your results. This could also strengthen the discussion, as it is quite common to refer to findings from those studies relative to the current study findings in the discussion and conclusions sections.

Response from the (co-)authors:

Thanks for this comment, we have added more comparisons in the discussion section between our findings and the relevant findings in literature.

Comment (3):

Conclusions must be drawn. These should contain implications for further research and derive recommendations for action.

Please add a table with the most important findings.

Several Limitations should be listed.

Response from the (co-)authors:

Thanks for this comment. In the revised manuscript, we expand the original discussion section to a new section, named as “Conclusion & Discussion”, in which we first list three most important findings. Following that we also add some discussion on the interpretation of these findings and their policy implication. In the end, we discuss the limitation of this paper and its potential extension in the future.

Finally, we sincerely appreciate the valuable comments by the reviewer, which help a lot in improving the quality of our paper.

Reviewer 3 Report

Line 50-51: a reference for domestic violence is also suggested

Line 55-58: “As the lockdown is always a hysteresis response to the severe outbreak of pandemic (Qiu et al., 2020; Zhang et al., 2020), while the end of lockdown is also accompanied with the end of pandemic, it is not 57 easy to disentangle the effect of lockdown from the effect of the pandemic itself.” As this is one of the key concepts of the paper, I suggest to expand this theme in the Introduction section, highlighting the complexity of underlying factors and of their interaction (e.g., fear of COVID, fear of contagion, hope, loneliness, uncertainty, and so on). As an example, studies predicting the psychological distress related to the COVID-19 indicated that during the first wave of COVID-19 living with others was a risk factor for higher psychological distress, likely in the relation to the fear of contagion (doi: 10.3390/jcm9103350); on the other hand, research has also shown that hope has higher protective role on the psychological distress related to the COVID-19 pandemic for people living with others, indicating the protective role of social connectedness (doi: 10.3390/ejihpe13010005).

Line 104-106: “We shall apply the event study approach and the DID regression analysis to quantify the comparison analysis, which will help filter the impact of lockdown out of total effect of COVID-19 depression”. I am not sure Introduction is the best place for this sentence. I would shift it to the Method section.

Line 109-136: I think those paragraphs should be shifted in the Introduction section, if the authors think they are relevant for the paper

Line 149-152: On which basis did the authors chose the keywords?

Line 272-274: it is not clear. Suggest to rephrase more precisely and clearly.

Line 287-289: “This observation is consistent with the intuition that as the epicenter, the severe shock of  COVID-19 epidemic crisis on Wuhan and its incurred lockdown did generate significantly negative impact on residents’ mental health which further promoted the conflict among family members and increased the search for the topic of domestic violence.” This sentence is more about discussion than about results.

Line 295-299: Same consideration that for line 287-289.

Line 308-311: Same consideration that for line 287-289.

Same consideration I suggest for other paragraphs in the Results section, where the authors hypothesize explanation/interpretation of their results.

Discussion: discussion should be expanded. The authors should make reference to previous studies and support their consideration based on previous literature.

Author Response

Dear reviewer,

Thank you for your comments on our manuscript entitled “Psychological impact of COVID-19 lockdown and its evolution: A case study based on internet searching data during the lockdown of Wuhan 2020 and Shanghai 2022” (ID: healthcare-2120478). Those comments are very helpful for revising and improving our paper. We have studied the comments carefully and made corrections. The main corrections are in the manuscript and the responds to the reviewers’ comments are as follows (the replies are highlighted with green color).

Comment (1):

Line 50-51: a reference for domestic violence is also suggested

Response from the (co-)authors:

We have added to reference in the suggested place.

Comment (2):

Line 55-58: “As the lockdown is always a hysteresis response to the severe outbreak of pandemic (Qiu et al., 2020; Zhang et al., 2020), while the end of lockdown is also accompanied with the end of pandemic, it is not 57 easy to disentangle the effect of lockdown from the effect of the pandemic itself.” As this is one of the key concepts of the paper, I suggest to expand this theme in the Introduction section, highlighting the complexity of underlying factors and of their interaction (e.g., fear of COVID, fear of contagion, hope, loneliness, uncertainty, and so on). As an example, studies predicting the psychological distress related to the COVID-19 indicated that during the first wave of COVID-19 living with others was a risk factor for higher psychological distress, likely in the relation to the fear of contagion (doi: 10.3390/jcm9103350); on the other hand, research has also shown that hope has higher protective role on the psychological distress related to the COVID-19 pandemic for people living with others, indicating the protective role of social connectedness (doi: 10.3390/ejihpe13010005).

Response from the (co-)authors:

Thanks for the valuable suggestion of the relevant literature. We have added the two references and the related discussion in the introduction section.

Comment (3):

Line 104-106: “We shall apply the event study approach and the DID regression analysis to quantify the comparison analysis, which will help filter the impact of lockdown out of total effect of COVID-19 depression”. I am not sure Introduction is the best place for this sentence. I would shift it to the Method section.

Response from the (co-)authors:

Thanks for this revision proposal, we have moved this sentence to the beginning of the method section.

Comment (4):

Line 109-136: I think those paragraphs should be shifted in the Introduction section, if the authors think they are relevant for the paper

Response from the (co-)authors:

Thanks for this revision proposal, we have followed the suggestion of the reviewer and put this background discussion of the data into the introduction section.

Comment (5):

Line 149-152: On which basis did the authors chose the keywords?

Response from the (co-)authors:

Comment (6):

Line 272-274: it is not clear. Suggest to rephrase more precisely and clearly.

Response from the (co-)authors:

Thanks for this comment, we have made some clarification on the mentioned sentence and given a clearer description on what the figure demonstrates.

Comment (7):

Line 287-289: “This observation is consistent with the intuition that as the epicenter, the severe shock of  COVID-19 epidemic crisis on Wuhan and its incurred lockdown did generate significantly negative impact on residents’ mental health which further promoted the conflict among family members and increased the search for the topic of domestic violence.” This sentence is more about discussion than about results. 

Line 295-299: Same consideration that for line 287-289.

Line 308-311: Same consideration that for line 287-289.

Same consideration I suggest for other paragraphs in the Results section, where the authors hypothesize explanation/interpretation of their results.

Response from the (co-)authors:

Thanks for this revision proposal. However, we do not quite agree with this comment. In line 295-299, except the last sentence “the observation above suggests that the lockdown, instead of the sharply increased infectious cases, should be the real source of the increasingly searching for domestic violence”, all the remaining part refers to the data description, instead of the discussion or interpretation.

All the four lines talked about one thing that the time point when the searching intensity on domestic violence related started to increase does not match the time point when the infectious cases start to increase. From this mismatch, we draw a conclusion that this observation suggests that the execution of lockdown is more likely than the spreading of COVID-19 to be the source of the impact on mental health.

Based on the content, we believe line 295-299 had better be placed in the original place, rather than move into the discussion.

As for the other lines, we agree with the reviewer that the discussion and interpretation should be separately reported with the results. But a strict separation may not be good for understanding. In fact, the content of the current study is not about a strict psychological experiment. It is based on observational data, instead of experimental data. For observational data, we cannot control the data generation process to let the data itself tell all the facts. When the data is insufficient to tell all the story, we need some explanation/interpretation on the results, which can assist us to understand the results. For this purpose, we believe add the minimal comment in aligned with the report of results can help the reader to better understand. It would be more efficient than to leave the “pure” results to the audience. On the other hand, we do need to take more care in the revised version and put as less subjective inference as possible in the result section. To this goal, we have rephrased line 287-289, line 295-299, line 308-311, and kept the comments from the author’s view to the least.

Comment (8):

Discussion: discussion should be expanded. The authors should make reference to previous studies and support their consideration based on previous literature.

Response from the (co-)authors:

Thanks for this revision proposal, we have expanded the current discussion section and move the mentioned paragraphs into the new discussion section. Meanwhile, we also add 6 more references that link the aforementioned discussion contents with the literature.

Finally, we sincerely appreciate the valuable comments by the reviewer, which help a lot in improving the quality of our paper.

Round 2

Reviewer 2 Report

The authors improved their paper, thanks.

Reviewer 3 Report

The changes improved the quality of the manuscript.

The authors could provide a further expansion of the discussion section, to further strenghthen the value of the paper.